# Chemical Analysis, Toxicity Study, and Free-Radical Scavenging and Iron-Binding Assays Involving Coffee (*Coffea arabica*) Extracts

**DOI:** 10.3390/molecules26144169

**Published:** 2021-07-08

**Authors:** Nuntouchaporn Hutachok, Pimpisid Koonyosying, Tanachai Pankasemsuk, Pongsak Angkasith, Chaiwat Chumpun, Suthat Fucharoen, Somdet Srichairatanakool

**Affiliations:** 1Department of Biochemistry, Faculty of Medicine, Chiang Mai University, Chiang Mai 50200, Thailand; n.hutachok@hotmail.com (N.H.); pimpisid_m@hotmail.com (P.K.); 2Department of Plant and Soil Sciences, Faculty of Agriculture, Chiang Mai University, Chiang Mai 50200, Thailand; tanachai.p@cmu.ac.th; 3Royal Project Foundation, Chiang Mai 50200, Thailand; pongsak.a@cmu.ac.th (P.A.); chdeech@yahoo.com (C.C.); 4Thalassemia Research Center, Institute of Molecular Biosciences, Salaya Campus, Mahidol University, Nakorn Pathom 70130, Thailand; suthat.fuc@mahidol.ac.th

**Keywords:** coffee, *Coffea arabica*, phenolic, free-radical scavenging, iron chelating, cytotoxic

## Abstract

We aimed to analyze the chemical compositions in Arabica coffee bean extracts, assess the relevant antioxidant and iron-chelating activities in coffee extracts and instant coffee, and evaluate the toxicity in roasted coffee. Coffee beans were extracted using boiling, drip-filtered and espresso brewing methods. Certain phenolics were investigated including trigonelline, caffeic acid and their derivatives, gallic acid, epicatechin, chlorogenic acid (CGA) and their derivatives, *p*-coumaroylquinic acid, *p*-coumaroyl glucoside, the rutin and syringic acid that exist in green and roasted coffee extracts, along with dimethoxycinnamic acid, caffeoylarbutin and cymaroside that may be present in green coffee bean extracts. Different phytochemicals were also detected in all of the coffee extracts. Roasted coffee extracts and instant coffees exhibited free-radical scavenging properties in a dose-dependent manner, for which drip coffee was observed to be the most effective (*p* < 0.05). All coffee extracts, instant coffee varieties and CGA could effectively bind ferric ion in a concentration-dependent manner resulting in an iron-bound complex. Roasted coffee extracts were neither toxic to normal mononuclear cells nor breast cancer cells. The findings indicate that phenolics, particularly CGA, could effectively contribute to the iron-chelating and free-radical scavenging properties observed in coffee brews. Thus, coffee may possess high pharmacological value and could be utilized as a health beverage.

## 1. Introduction

*Coffea arabica* (Arabica) and *Coffea acanephora* (Robusta) are known to be two of the most popular beverages in the world; however, Arabica coffee is more often consumed and more preferable in the global coffee market [1,2]. Coffee cherry husks, as well as green and roasted coffee beans, have all been processed to produce popular coffee beverages, of which roasted coffee beans are recognized as the most popular. Green coffee extract is made of unroasted green coffee beans. It is available as a dietary supplement and contains phenolic amides, as well as other phytochemicals [3,4]. In fact, the physical aspects, the species of the coffee bean, and the roasting and brewing processes are all important factors that influence the chemical composition of coffee beverages [5]. These chemical compositions include caffeine (CF), chlorogenic acid (CGA), caffeic acid (CA) and Maillard reaction products (e.g., melanoidins) [1,5]. Besides providing an alerting effect, coffee consumption is also associated with a range of health complications such as insomnia, tremors, nausea, polyuria, diarrhea, polyphagia, hypertension and a decrease in iron absorption. All of which have been attributed to the CF and melanoidins content in coffee-based beverages [6,7,8,9]. Beneficially, coffee intake can reduce the risks associated with type 2 diabetes mellitus, Parkinson’s disease, colorectal cancer, hepatic injury, cirrhosis and hepatocellular carcinoma [10,11,12,13,14,15]. These benefits have been attributed to the actions of nitrogenous compounds, acids, esters and CGA [5,16,17,18]. CF (1,3,7-trimethylxanthine) is naturally found in coffee beans, cacao beans, kola nuts, guarana berries and tea leaves, of which coffee and tea are the first and second most prominent sources. The performance benefits of CF include the enhancement of mental alertness, increased levels of concentration and physical endurance, a potential reduction in fatigue and body weight and a lowering of the overall risks associated with certain metabolic syndromes [19,20]. CGA has three subclasses including5-*O*-caffeoylquinic acid (CQA), feruloylquinic acid (FQA) and dicaffeoylquinic acid (diCQA), of which CQA is the most common and strongest antioxidant present in coffee in the form of neochlorogenic acid (3-CQA), cryptochlorogenic acid (4-CQA) and chlorogenic acid (5-CQA) [21].

Instant coffee (regular and decaffeinated type) is a spray-dry form of coffee made from coffee extracts combined with a number of other functional ingredients (e.g., vitamins A and C, iron, inulin and oligofructose); nonetheless, the fortification of instant coffee products is necessary to improve particle size distribution, reconstitution properties, wettability and dispersibility times, as well as the overall level of satisfaction of its consumers [22]. In the manufacturing process, decaffeinated coffee is usually prepared by a treatment with waterand an organic solvent or carbon dioxide to remove intact CF from coffee beans before they are roasted and ground.The resulting coffee beveragewill then contain 1–2% of the original CF content of the regular coffee. Even after removing CF, the decaffeinated type still contains phenolic compounds (such as CGA, CA and trigonelline) and other phytochemicals. Thesecompounds and contents are similar to those found in the regular type and are known to exertcertain biological activities of interest [23,24,25,26,27].

Depending on the country of origin and the differing preferences of cultures and individual, coffee can be brewed by simple percolation or boiling methods, or with the use of Italian and electric coffee makers, espresso machines and French presses [5]. However, instant coffee is made by drying prepared coffee which produces a soluble powder that can be dissolved in hot water by the consumer. Different coffee preparations result in different tastes, aromas and chemical compositions [5,9,28,29,30]. It is likely that changes in the phenolics and CGA contents in coffee brews can affect these biological activities [30,31]. In contrast, degradation of 5-CQA that occurs during the roasting process does not affect antioxidant activity, whereas higher CF content has resulted in a greater degree of antioxidant activity indicating that the antioxidant activity may not depend only upon the CGA action [32].

Many liquid chromatographic techniques have been developed for identification of the active ingredients in coffee samples. For instance, the high-performance liquid chromatography-diode array detection (HPLC-DAD) method is often used for the simultaneous quantification of CF, trigonelline, nicotinic acid, N-methylpyridinium ion, 5-CQA, and 5-hydroxymethyl furfural. The resulting values can then be compared to the specific retention times (T_R_) and concentrations of the authentic standards [33]. HPLC coupled witha mass spectrometer and a nuclear magnetic resonance spectrometer has been developed for the efficient analysis of the phenolic compounds in coffee bean extracts [34]. Recently, a fast highly-resolved sensitive ultra-high-performance liquid chromatography coupled with electrospray ionization quadrupole time-of-flight mass spectrometry (UHPLC-ESI-Q-TOF-MS), that involves a greater degree of informative structure elucidation and identification of the fragmentation patterns of the compounds, has been developed to identify polyphenols, alkaloids, diazines, and Maillard reaction products present in ground coffee samples [35]. Furthermore, a fast direct form of analysis using a real time ion source coupled with high-resolution time-of-flight mass spectrometry (DART-TOF-MS) without any prior chromatographic separationhas been developed for the quantitative analysis of CF in coffee samples [36]. In most advanced research studies, this technique can be applied for pharmaceutical, phytochemical and metabolomic analysis [37,38]. The present study aims to analyze the chemical compositions, assess the free-radical scavenging and iron-chelating activities and evaluate the toxicity of different coffee preparations.

## 2. Results

### 2.1. Information and Extraction Yield of Coffee Samples

Table 1 summarizes types and preparations of coffee samples used in this study, in which details and preparation protocols have been explained in the Materials and Methods section.

Yields for the extracts of ground roasted coffee beans prepared from boiling and with the use of automatic coffee makers (drip and espresso methods) were found to be 17.76, 16.04 and 9.51% (*w/w*), respectively. The coffee extracts were further analyzed in terms of their chemical composition using the more sensitive HPLC-ESI-MS, UHPLC-ESI-Q-TOF-MS, while CA, CF and CGA concentrations were determined using HPLC/DAD. The antioxidant and iron-binding activities of the coffee products were determined and the degree of toxicity in cells and animals was evaluated.

### 2.2. HPLC-ESI-MS Identification of Phenolic Compounds in Coffee Bean Extracts

The phenolic constituents present in green and roasted coffee bean extracts were then analyzed using high-resolution HPLC-ESI-MS. Table 1 presents the chemical characterization of all identified phenolic compounds by peak elution order: T_R_, UV absorption maxima at 270 nm from adiode array detector (DAD), exact molecular mass, molecular formula, quasimolecular ions ([M-H]^+^, [M-NH_4_]^+^, [M-Na]^+^ and [M-K]^+^) with relative abundance and tentative names. As is shown in Figure 1A–F and Table 2 all analyzed roasted and green coffee extracts displayed nearly the same qualitative profiles of the bioactive substances found in the HPLC/DAD profiles at 270 nm during the course of monitoring the phenolic compounds. Their peaks were identified on the basis of UV spectra and elution/retention sequences that had been reported in previously published literature. These values were then confirmed by their mass spectrometric behavior. At least 17 phenolic compounds were detected in all roasted coffee extracts including trigonelline (peak 2), caffeic acid (peak 3), gallic acid (peak 4), epicatechin (peak 5), dicaffeoylquinic acid or dichlorogenic acid (peak 6), caffeoylquinic acid or chlorogenic acid (peak 7), caffeoyl-*O*-hexoside (peak 8), *p*-coumaroylquinic acid (peak 9), *p*-coumaroylglucoside (peak 10), rutin (peak 11), caffeoylquinic acid or chlorogenic acid derivative (peak 12), syringic acid (peak 13), caffeoylquinoyl-*O*-glucoside or chlorogenoyl-*O*-glucoside (peak 14), *p*-caffeoylquinoyldiglucoside I or chlorogenoyldiglucoside I (peak 15), *p*-caffeoylquinoyldiglucoside II or chlorogenoyldiglucoside II (peak 16), dicaffeoylquinoyl-*O*-glucoside or dichlorogenoyl-*O*-glucoside(peak 17) and an unidentified compound (peak 1) (Figure 1A–C and Table 1). In comparison, boiled and drip green coffee extracts contained the same 17 phenolic compounds and two additional phenolic compounds, namely dimethoxycinnamic acid (peak 10a) and caffeoylarbutin (peak 11a) (Figure 1D,E and Table 1), while espresso green coffee extract contained the same 17 phenolic compounds and three additional phenolic compounds, namely dimethoxycinnamic acid (peak 10a), caffeoylarbutin (peak 11a) and cynaroside (peak 13a) (Figure 1F and Table 1). Hence, there were no differences in the chromatographic profiles of the phenolic compounds analyzed in roasted and green coffee extracts that were prepared using the boiling, drip and espresso methods, with the exceptionof the aforementioned compounds.

### 2.3. UHPLC-ESI-Q-TOF-MS Identification of Phytochemical Compounds

According to the UHPLC-Q-TOF-MS analysis, many phytochemical compounds including 4-fluoro-L-threonine (peak 1), 3-nitroperylene (peak 2), cycloeudesmanesesquiterpenoids (peak 3), 3,4,5-tricaffeoylquinic acids (peak 4), 6-gingesulfonic acid (peak 5), phytosphingosine (peak 6), sativanine B (peak 8), sterebin E (peak 11), anofinic acid (peak 12), samandenone (alkaloids) (peak 13), sorbitan oleate (peak 14), 2-palmitoylglycerol (peak 15), citranaxanthin (peak 16), 2-stearoyl glycerol (peak 17), dodecanic acid and 12-methoxy-1-[(phosphonooxxy)methyl]1,2-ethanediyl ester (peaks 18, 19) were detected in extracts of boiled, drip and espresso roasted coffee beans and the extracts of boiled, drip and espresso green coffee beans (Figure 2 and Table 3). In addition, citronellyl butyrate, 2-[2-(4-hydroxy-3-meyhoxyphenyl)ethyl]tetrahydro-6-(4,5-dihydroxy-3methoxyphenyl)-2H-pyran-4-ylacetateand dodemorph, which correspond to peaks 7, 9 and 10, respectively, weredetected only in the boiled roasted coffee extract. However, the compounds eluted at T_R_ of 34.50 min (peak 20) and 36.65 min (peak 21) are unknown. The results imply that not only were two major active compounds, namely CF and CGA, found in the green and roasted Arabica coffee beans, but also that a number of minor phytochemical compounds were found in the beans. Owing to the presence of certain nutrients or the function of other important biological/pharmacological properties, these coffee extracts require further investigation.

### 2.4. CF, CGA and Total Phenolic Contents

HPLC-DAD profiles shown in Figure 3 have demonstrated the presence of CF and CGA, but not CA in these coffee extract samples. Stoichiometric data have revealed that the CGA contents were equal among the coffee extracts, while CF contents of the boiled and espresso coffee extracts were higher than that of the drip coffee extract. Inversely, TPC of the drip coffee extracts was higher than those of the boiled and espresso coffee extracts (Table 4). Herein, regular and decaffeinated instant coffee types were found to contain equal amounts of TPC, while their CA, CF and CGA contents were not determined in this study.

### 2.5. Free-Radical Scavenging Activity

In this study, the antioxidant activities of coffee samples were determined using ABTS^+●^ and DPPH^●^ methods in which Trolox was used as a standard for both experiments. Considerably, the results demonstrated that all coffee extracts and instant coffee preparations expressed free-radical scavenging properties in a concentration-dependent manner. With regard to the Trolox equivalent (TE), anti-oxidation values were assayed using the ABTS method, wherein the drip coffee extract (149.4 ± 9.2 mg TE/g) was found to be significantly more effective than the boiled coffee extract (125.8 ± 9.1 mg TE/g) and the espresso coffee extract (127.6 ± 3.0 mg TE/g) *(p* < 0.05) (Figure 4A). In this regard, the regular instant coffee (160.4 ± 4.0 mg TE/g) seemed to be as effective as the decaffeinated instant coffee (173.2 ± 8.9 mg TE/g) (*p* > 0.05).

Similarly, Trolox as well as the coffee samples were able to scavenge DPPH^●^ in concentration-dependent manners; however, the abilities were significantly enhanced when consecutive concentrations were increased (Figure 4B). In addition, the DPPH^●^ scavenging activities of the coffee samples are depicted as TE values (mean ± SD). Therefore, the espresso coffee extract (1274 ± 46 mg TE/g) and the drip coffee extract (1250 ± 38 mg TE/g) could exert stronger antioxidant activity than the boiled coffee extract (1174 ± 26 mg/g extract) (*p* < 0.05). However, regular instant coffee (2359 ± 159 mg/g) exhibited antioxidant activity equal to decaffeinated instant coffee (2358 ± 93 mg/g). The antioxidant activity of the coffee extracts and instant coffee were assayed using the ABTS^+^^●^ and DPPH^●^ methods and the results are summarized in Table 5.

These findings have indicated that two instant coffee products displayed significantly stronger radical-scavenging activity than all three coffee extracts in the following order: drip coffee extract > espresso coffee extract and boiled coffee extract based on ABTS assay and drip coffee extract = espresso coffee extract >boiled coffee extract based on DPPH assay.

### 2.6. Iron-Binding Ability

Using scanning spectrophotometry, all the coffee samples themselves elucidated three distinct maximal absorption peaks at wavelengths of 217, 275 and 323 nm in a concentration-dependent manner (Figure 5A–C, Table 6). However, the decaffeinated instant coffee resulted in a longer absorption wavelength at 289 nm than the regular instant coffee (Figure 5D,E, Table 6), while CF itself exhibited two major absorption peaks at 217 and 275 nm and CGA itself exhibited peaks at 217 and 324 nm (Figure 5F,G, Table 6). Additionally, no peaks were exhibited by all of the samples at wavelengths of 400–600 nm.

In comparison, CGA, but not CF, was found to bind ferric ion from Fe-NTA in a concentration-dependent manner producing a complex with a maximal absorption peak of 617 nm (Figure 5M,N, Table 6). Notably, all coffee extracts and instant coffee varieties also bound ferric ion in a concentration-dependent manner to produce a complex with a new maximal absorption peak of 595 nm (Figure 5H–L, Table 6).

### 2.7. Cytotoxic Effects

Herein, Arabica coffee extracts prepared by boiling, drip and espresso methods were tested in vitro for toxicity against normal human PBMC and human breast cancer (MDA-MB-231 and MCF-7) cells within 24 and 48 h of incubation and the results are shown in Figure 6. PBMC viability was found to be almost unchanged following treatment with all coffee extracts (with the exception of the 25 and 50 µg/mL espresso coffee beverages, (*p* < 0.05) for 24 h and compared to the specimens without treatment. During 48h of incubation, cell viabilities were significantly increased in a concentration-dependent manner by all the extracts, for which the drip coffee beverage was the most effective when compared to the specimens without treatment. In comparison, the viability of MDA-MB-231 cells remained unchanged following treatment with all the extracts for 24 and 48 h. When MDA-MB-231 and MCF-7 cells were treated with aggressive doses of the drip coffee extract at 31.25–1000 µg/mL, the viability of these two cells was unchanged during incubation for 24 h and tended to decrease slightly during the course of incubation up to 48 h. According to the findings, all three coffee extracts were non-toxic to normal PBMC but expressed a significantly dose-dependent increase in cell viability during 48 h of incubation based on the reducing power of the mitochondrial reductases. Likewise, no cytotoxic effect was observed in MDA-MB-231 cancer cells treated with equal concentrations of the coffee extracts for 24 and 48 h. Moreover, increasing doses of the drip coffee extract and treatment times up to 48 h were determined to be not harmful to both MDA-MB-231 and MCF-7 cells.

### 2.8. Acute Oral Toxicity in Rats

Acute toxicity test was conducted following OECD Guidelines for the Testing of Chemicals 425 by using initial doses of the coffee extracts at 175, 550 and 2000 mg/kg body weight (BW), respectively. However, the upper limit of the caffeine dose should not exceed 400 mg/kg BW. All clinical signs and symptoms were observed and recoded in Table 7. At the first and second dose of the drip coffee extract, neither a toxic effect nor a lethal effect was observed in the rats. One out of three rats dosed at 2000 mg/kg exhibited a sign of lethargy and drowsiness after 1 h of being given a dose of the coffee extract. After 6 h and 14 days of observation, no clinical signs of toxicity and lethality were observed. Since there were no signs of mortality and no clinical signs of toxicity at all dosing levels, the median lethal dose (LD_50_) value of the drip coffee extract was found to be greater than 2000 mg/kg.

## 3. Discussion

Coffee is understood to be the most frequently consumed beverage in most countries worldwide. The consumption of coffee can be beneficial or may actually be detrimental to human health due to the naturally occurring active compounds that are present in coffee products. Though diterpenes (e.g., cafestol and kahweol), phenolics (e.g., CGA, CA) and heterocyclic compounds present in boiled coffee (e.g., CF and melanoidins) exhibit strong antioxidant properties, these two diterpenes have been claimed to be associated with increased serum cholesterol levels [39,40]. In the coffee trade, coffee beans are extensively used in beverage processing and are mainly comprised of a number of functional phytochemicals and nutritional carbohydrates [41]. Certain extraction factors can influence the quantity and activities of bioactive compounds that are present in coffee samples. These extraction factors include the use of solvents, the mass to volume ratio, acidity, time, temperature and pressure, as well as the application of microwaves or ultrasonic preparation methods and the specific type of coffee maker that may have been used [42,43,44,45]. Brewing is a key and final step in the production process of coffee drinks. The resulting drinks have been associated with coffee stoichiometry (known as total dissolved solids (TDS)), percent extraction and sweetness, and an inverse proportion to TDS. In addition, roasting is another key step in the preparation process of many of the most popular brewed coffee beverages. The roasting step can deliver a pleasant aroma while minimizing the bitterness of the final coffee beverage [46]. In this study, we have emphasized that all data reported were based on the use of a single-type of wet-washed Arabica coffee beans to prepare coffee beverages with and without roasting. This coffee bean was selected for the production of brewed coffee using boiled water and automatic coffee makers (drip and espresso types). Thus, we have demonstrated higher coffee extraction yields by using the boiled water and drip methods over the espresso method. A previous study reported the TDS values of the espresso and drip brewed coffees (5–10% and 1.0–1.75%, respectively), and the extraction yields of the espresso coffee (approximately 14–25%) [47]. In comparison, the extraction yields of the commercial coffee samples were found to be 81.26 ± 19.23 mg/g by methanol (100 °C), 19.12 ± 1.28 mg/g by dichloromethane (120 °C) and 1.76 ± 0.93 mg/g by *n*-hexane (120 °C) [48]. Consistent with the outcomes of our study, coffee drinks brewed with high temperatures and the cold dripping method exhibited the highest values in terms of TDS, extraction yields and the highest contents of CF, trigonelline, 4-CQA and 5-CQA, regardless of the roasting method that was used [49]. Moreover, Angelone and coworkers have reported on the extraction yield values assayed in espresso (13.1 ± 1.3–22.8 ± 1.3%), moka (28.4 ± 1.1%), V60 (22.1 ± 0.7%), cold brew (23.3 ± 0.9%), Aeropress (20.4 ± 1.2%) and French press (18.7 ± 1.1%) coffee beverages [50]. In roasting, galactomannan yields extracted from ground coffee with hot water were observed to increase [51]. Importantly, the amount of ochratoxin A, a naturally occurring food-borne mycotoxin produced by *Aspergillus* spp. and *Penicillium* spp. that is often found in green, roasted and brewed coffee products was reduced by the use of an espresso coffee maker (49.8%), the drip-filter method (14.5%) or the moka brewing process (32.1%) [52]. In terms of roasting coffee, CF content was not affected, CGA was degraded due to a consequence of the temperature used in brewing, CA decreased in dark roasted coffee, while melanoidins and other Maillard reaction products were developed [53]. Importantly, the roasting of green coffee beans and the brewing of ground roasted coffee are important processes that are used to make varieties of bioactive, aromatic and popular coffee drinks.

Ultraviolet (UV) detection, derivatization spectrophotometry and gas-liquid chromatography (GLC) techniques have been developed for the simple and rapid determination of CF concentrations in beverages over a long period of time [54]. At present, HPLC-ESI-MS and HPLC-ESI-MS-MS are known to be powerful techniques with high degrees of sensitivity and accuracy that can be used to determine the compound profiles of plant materials and natural products. Owing to the advantages of MS detection, coffee extracts were subjected to HPLC-ESI-MS identification by collision-induced dissociation mass spectrometer in order to identify certain phenolics, especially isomers. MS fragmentation patterns were observed after analysis by tandem MS spectra, while chromatographic retention times, relative hydrophobicity and bonding strength to quinic acid have been used to develop structure-diagnostic hierarchical keys for the identification of phenolic compounds. In this sense, the relative hydrophobicity of aglyconic phenolics can depend upon the substitution position of the phenolic ring and the number and identity of the residues. By using HPLC/ESI-MS/MS, we have identified at least 17 phenolic compounds, including trigonelline, CA, gallic acid, epicatechin, di-CGA, CGA, caffeoyl-*O*-hexoside, *p*-coumaroylquinic acid, *p*-coumaroylglucoside, rutin, CGA derivative, syringic acid, CGA-glucoside, CGA-diglucoside I, CGA-diglucoside II, diCGA-glucoside, and one unknown compound in the roasted coffee extracts, whereas all of the 17 compounds and an additional three compounds, namely dimethoxycinnamic acid, caffeoylarbutin and cynaroside, were detected in the green coffee extracts. Interestingly, trigonelline, which is a bitter alkaloidal ingredient that serves as an aroma generator and is responsible for certain biological activities, was detected in both green and roasted coffee extracts. Stennert and Maier demonstrated that trigonelline was degraded gradually during the roasting stage [55]. Similarly, CQAs, including 5-CQA,3-CQA and 4-CQA; diCQAs including 3,4-diCQA, 3,5-diCQA and 4,5-diCQA; feruloylquinic acids (FQAs) including 3-FQA, 4-FQA and 5-FQA; diFQA and *p*-coumaroylquinic acids (*p*-CoQAs) including 3-*p*-CoQA, 4-*p*-CoQA and 5-*p*-CoQA isomers, were also found to be present inthe coffee samples [56]. Likewise, CF, CQAs, diCQAs, *p*-CoQAs, FQAs and caffeoylquinic acid lactone were detected in the espresso, moka, cold brewed and French Press coffee extracts [50]. This technique has been reported to be effective in identifying certain bitter compounds, such as 1,3-bis(3′,4′-dihydroxyphenyl) butane, trans-1,3-bis(3′,4′-dihydroxyphenyl)-1-butene and eight hydroxylated phenylindanes in roasted coffee at threshold concentrations of 23–178 μM [57], as well as in the detection of a carcinogenic furan in defatted, ground and constituted coffee preparations [58]. In the roasting process, free amino acids and peptides existing in green coffee beans are changed into aromatic flavors, whereas polymerization and fragmentation of proteins simultaneously generate hydrogen peroxide [59]. Unfortunately, a potential carcinogen acrylamide was also detectable in certain foods including coffee (169 ng/g) [60].

In addition, HPLC can be employed with ESI-triple quadrupole time-of-flight mass spectrometry (HPLC-ESI-QTOF-MS) in order to provide higher resolution, faster speeds and less solvent consumption, which can lead to a rapid and sensitive characterization of certain unexpected natural products. Currently, Spreng and colleagues applied this technique for the analysis of roasted coffee and reported eleven pyrazine derivatives, of which 2-(2′,3′,4′-trihydroxybutyl)-(5/6)-methyl-pyrazine and 2,(5/6)-bis(2′,3′,4′-trihydroxybutyl)-pyrazine were the most prominent compounds [61]. In the new findings, our UHPLC-ESI-QTOF-MS results have elucidated the presence of many phytochemicals, including 4-fluoro-L-threonine, 3-nitroperylene, cycloeudesmanesesquiterpenoids, 3,4,5-tri-CQAs, 6-gingesulfonic acid, phytosphingosine, sativanine B, sterebin E, anofinic acid, samandenone, sorbitan oleate, 2-palmitoylglycerol, citranaxanthin, 2-stearoyl glycerol, dodecanic acid and 12-methoxy-1-[(phosphonooxxy)methyl]1,2-ethanediyl ester, citronellyl butyrate, 2-[2-(4-hydroxy-3-meyhoxyphenyl)ethyl]tetrahydro-6-(4,5-dihydroxy-3-methoxyphenyl)-2H-pyran-4-ylacetate and dodemorph, in the extracts of the boiled, drip and espressogreen/roasted coffee beans. In addition, UHPLC-ESI-QTOF-MS analysis of the clinical specimens obtained from selected coffee consumers has indicated the appearance of CF, methylxanthines and methyluric acids in the plasma, along with eleven methylxanthine and methyluricacid metabolites, furan and methylfuran metabolites in the urine. These were mainly found in the form of sulfate, methyl derivatives, glucuronides and un-metabolized CQAs, FQAs, CoQAs, CA, ferulic acid and coumaric acids [62,63,64].

Through the use of HPLC-DAD analysis, we have detected the presence of CF and CGA, but not CA, in the roasted Arabica coffee extracts that were prepared by boiling and with the use of coffee makers (drip and espresso) [65]. A recent HPLC/DAD analysis of five commercial coffee samples has reported the presence of 8.35 ± 6.13 mg CF/g methanol extract [48]. Furthermore, trigonelline, CF and CGA were detected in green coffee beans (0.65 ± 0.05, 0.97 ± 0.09 and 3.13 ± 0.33 mg/g dry weight, respectively) and roasted coffee beans (0.85 ± 0.01, 1.30 ± 0.13 and 1.00 ± 0.02 mg/g dry weight, respectively), indicating a decrease in CGA and total phytochemical contents but increases in trigonelline and CF contents during the roasting process [66]. Likewise, trigonelline, CGA and CA were found in instant coffee samples [24]. Due to the thermal degradation of CGA and the relative stability of the alkaloids that occur at high temperatures during the roasting process, CF and trigonelline were the main metabolites present in roasted coffee beans [66]. During the process of roasting, CGAs and their CGA derivatives in green coffee that contribute to the acidity, astringency and bitterness of the final coffee beverages, can be isomerized and transformed to CGA lactones through a process involving water loss from quinic acid moiety and intramolecular ester bonding. This gives the coffee its flavor and determines its quality [67]. However, the amount of CGAs and their derivatives can also be used to indicate the quality of green coffee beans and to discriminate between low-quality varieties (9.1 g CGA/100 g) and commercial ones (10.4 g/100 g) [68]. Consistently, the phenolics found in our hot water/drip coffee extracts were mostly similar to those found in cold drip coffee extracts as determined by Angeloni et al. [69]. Through the use of HPLC/MS in our study, the phenolic and alkaloid profiles identified for the espresso coffee extracts were not totally the same as those assayed by Aves and colleagues [70]. However, the analyzed compounds were mostly comprised of trigonelline, CQAs, FQAs, coumaroylquinic acids, diCGAs, diFQA and caffeoylferuloylquinic acid. In the present study, when using UHPLC-ESI-QTOF-MS, 19 of 21 phytochemical compounds were detected consistently in all of the green coffee and roasted coffee extracts. Consistently, Angelino and colleagues used the UHPLC-MS-MS technique and have revealed the presence of trigonelline, CF, CGA, FQA, coumaroylquinic acids, hydroxycinnamate dimers, caffeoylshikimic acids and caffeoylquinic lactones in regular espresso coffee varieties [71].

With regard to antioxidant activity, food processing generally affects the content and activities of persisting phytochemicals and the corresponding antioxidant capacities in functional foods. Coffee beans are the main sources of anti-oxidative compounds andrequire roasting and drying before utilization. Consequently, the two processes may give rise to significant alterations in the antioxidant compositions and properties of theresultant coffee products, while the Strecker and Maillard reactions may increasefree-radical scavenging capacities [41]. Functional phenolic compounds contain hydroxyl components on the aglyconic phenolic ring and glyconic ring that scavenge ROS. Nonetheless, the antioxidant capacity that is related to human health benefits is dependent upon the bioavailability of the phytochemicals after consumption, which is subsequently dependent upon the soluble parts of the coffee known as the extraction yields [17]. In accordance with this finding, all roast coffee extracts and instant coffee doses dependently scavenged ABTS^●+^ and DPPH^●^ at different potencies depending on the coffee type and the preparation method used, in which the drip coffee extract exhibited the most efficient activity. With regard to the relevant technical factors, decaffeinated espresso coffee exerted slightly higher DPPH^●^ scavenging activity than regular espresso coffee (32% and 38%, respectively), which was directly related to the phenolic contents [70]. Iwai and colleagues previously elucidated that the potency order of superoxide anion radical scavenging activity was diCGA > CA, CGA > FQA, for which the activities of the diCGA were twice as effective as those of CGA and four times as effective as those of 5-FQA [68]. Importantly, total phenolic content, ABTS^●+^- and DPPH^●^-scavenging capacities of coffee samples were increased during the roasting process [66]. More importantly, the order of antioxidant capacity of all coffee brews was espresso > mocha > plunger > drip-filtered; herein, CGAs scavenged Fremy’s salt (potassium nitrosodisulfonate) radicals, while melanoidins scavenged 2,2,6,6-tetramethyl-1-piperidin-1-oxyl radicals [72].

Notably, the consumption of a cup of drip coffee or instant coffee resulted in approximately a 39% decrease in dietary non-heme iron. Additionally, when coffee is consumed along with a meal, ferric-ethylenediamine tetraacetic acid or ferric chloride decreases the degree of dietary iron absorption from 5.88% to 1.64% depending upon the coffee dose [6]. For the preparation of coffee extracts, water was found to be the optimum solvent in terms of producing higher yields, better protective effects against lipid peroxidation, and effective ROS-scavenging and iron-chelating properties when compared to methanol, ethanol and *n*-hexane [73]. It was noted that certain phenolic compounds, such as CGAs, hydrolysable multi-galloyl tannin and galloylcatechin, did not condense the tannins or catechins present in the coffee utilized diol groups onto the phenolic ring that would bind iron at different affinity levels and interfere with dietary iron absorption [74,75]. In addition, the CGAs abundant in roasted coffee beans utilized the hydroxyl groups to bind thiol-molecules [76]. Moreover, coffee grounds were found to adsorb divalent metal ions in the following order: Cu < Pb < Zn < Cd, whereas tea leaves exhibited a similar outcome in the opposite order [77].

It has been reported that coffee constituents (e.g., kahweol, cafestol and CGAs) induced antioxidant-response element gene expression and activated the production of anti-oxidative enzymes in peripheral blood lymphocytes [78]. Herein, our roasted coffee extracts significantly increased the MTT-assayed viability of normal PBMC at 48 h, but did not influence the viability of breast cancer MDA-MB-231 and MCF-7 cells at any time period. Consistently, the consumption of coffee containing cafestol and kahweol induced antioxidant enzymes by an increase of 38% of superoxide dismutase activity [79]. Taken together, the treatment of PBMC with kahweol, cafestol and CGAs-rich roasted coffee can increase and empower mitochondrial reducing enzyme systems and subsequently intensify blue formazan products, which can result in higher percentages of cell viability. Ambiguously, a recent placebo-controlled intervention trial in healthy subjects has demonstrated that the consumption of coffee at up to 5 cups per day had no detectable beneficial or harmful effects on human health [80]. In contrast, the biocompatible copper sulfate-oxidized nanoparticles of Arabica coffee bean extracts showed anti-proliferative activity against MCF-7 breast cancer cells [81]. In some studies, CGAs and the natural extracts of green and roasted coffee beans could be employed as chemoprotective dietary supplements against the proliferation of Ras-dependent breast cancer MDA-MB-231 cell lines [82]. On the other hand, trigonelline, a niacin-related natural constituent of coffee (1%), was found to stimulate MCF-7 cell growth by acting as a phytoestrogen through the mediation of the estrogen receptor [83]. However, there is no evidence to support a relationship of either caffeinated or decaffeinated coffee intake with breast cancer risk [84]. Despite the fact that they are the two major active flavor ingredients in coffee, CA (6 and 10 mg/plate) and CGA (19 and 28 mg/plate), have been reported for their mutagenic properties in the L5178Y mouse lymphoma TK^÷/-^assay. This was possibly due to the oxidative degradation and transformation of the two compounds to hydrogen peroxide at a neutral pH in the presence of a metal ion^2+^ (such as Mn^2+^) [85].Additionally, green and light roasted coffee extracts were found to promote higher inhibitory effects on the viability of PC-3 and DU-145 metastatic prostate cancer cell lines that had been assayed by the MTT test [53]. Moreover, green coffee beans containing CGAs and their derivatives showed anti-proliferative effects against U937, KB, MCF7 and WI38-VA cancer cell lines [68]. In controversy, diterpenes (e.g., cafestol and kahweol) are claimed to promote an increase in plasma cholesterol concentrations as a risk factor of cardiovascular diseases and type 2 diabetic mellitus [86].

## 4. Materials and Methods

### 4.1. Chemicals and Reagents

In terms of materials, 2,2′-azino-bis (3-ethylbenzothiazoline-6-sulfonic acid)diammonium salt (ABTS), chlorogenic acid (CGA), caffeic acid (CA), caffeine (CF), gallic acid (GA), 2,2-diphenyl-1-picrylhydrazyl (DPPH), formic acid, 3-(N-morpholino) propanesulfonic acid (MOPS), 3-[4,5-dimethylthiazol-2-yl]-2,5-diphenyl tetrazolium bromide (MTT), nitrilotriacetate disodium salt, nitrilotriacetate trisodium salt, phosphate-buffered saline (PBS) pH 7.0, phosphoric acid, 6-hydroxy-2,5,7,8-tetramethylchroman-2-carboxylic acid (Trolox) and potassium thiosulfate were purchased from Sigma-Aldrich, Chemical Company (St. Louis, MO, USA). Ferric nitrate nonahydrate, sodium carbonate and Folin-Ciocalteu’s phenol reagent were obtained from the Merck-Millipore Company (Merck KGaA, Damstadt, Germany). Organic solvents, such as acetonitrile, were all of HPLC grade and all chemicals were highly pure and of analytical grade. Commercial instant coffee (regular and decaffeinated types) were purchased from a Tesco Supermarket in Chiang Mai Province, Thailand.

### 4.2. Preparation of Coffee Extracts

Green coffee beans (*Coffea arabica*) were harvested from coffee fields belonging to the Royal Project Foundation at Mae Hair Village, Mae Wang District, Chiang Mai Province, Thailand. Whole beans were roasted and ground (known as milling) with the use of a coffee-grinding machine in a Coffee Factory of the Royal Project Foundation, Chiang Mai, Thailand. Extracts of ground green and roasted coffee were prepared using the boiled water, drip and espresso coffee maker methods, as has been previously described by Hutachok and colleagues [65]. Boiled coffee extract was manually prepared by placing ground roasted coffee (10 g) into hot water (90 °C) (100 mL), allowing the coffee to dissolve for 10 min and then allowing it to cool down. In pressurized percolation (espresso coffee extract), hot water (100 mL) was forced through fine coffee (10 g) under pressure using a manual portable Espresso maker (Minipresso GR, Wacaco Company, Pagewood NSW, Australia) with a specification of 8g ground coffee capacity, 120-mL water capacity and 8-bar pump pressure according to the manufacturer’s instructions. During the brewing of drip coffee, hot water (100 mL) was forced through fine coffee (10 g) and filtered on high-quality paper under pressure using an automated drip coffee maker (Delonghi, Thiptanaporn, Company Limited, Bangna, Bangkok, Thailand) with a specification of 1.4-L tank capacity and 15-bar pump pressure according to the manufacturer’s instructions. All the coffee brewed was filtered through a clean filter cloth and centrifuged at 6000 rpm for 20 min, while the supernatant was lyophilized to dryness. Coffee extracts were then stored in separate brown bottles at −20 °C until further analysis.

### 4.3. HPLC/ESI-MS Identification of Phenolic Compounds

All of the coffee extracts were analyzed in terms of their polar phenolic compounds at the Central Laboratory (North Branch), Department of Land Development, Ministry of Agriculture and Cooperation, Chiang Mai using the HPLC/MS method established by Cuyckens and colleagues with slight modifications [65,87]. The HPLC system (Agilent Technologies 1100 Series, Deutschland GmbH, Waldbronn, Germany) consisted of a quaternary pump (G1311A), an online vacuum degasser (G1322A), an autosampler (G1313A), a thermostated column compartment (G1316A) and a photodiode array (PDA) detector (G1315A). The outlet of the PDA was coupled directly to the atmospheric pressure ESI interface of the mass spectrometer (MS) detector (Agilent Technologies 1100 LC/MSD SL, Palo Alto, CA, USA) through a flow splitter (1:1). In terms of the analysis, coffee extracts were constituted in 1.0 mL of a mixture comprised of solvent A (acetonitrile) and solvent B (10 mM formate buffer pH 4.0) (1:1, *v/v*) and filtered through a syringe filter (polytetrafluoroethylene membrane, 25 mm diameter, 0.45-μm pore size, Corning^®^) before being used. The mixture was then injected (20 μL) into the system. Chromatographic separation was carried out on a column (LiChroCART RP-18e, 150 mm × 4.6 mm, 5 µm particle size; Purospher STAR, Merck, Darmstadt, Germany) operated at 40 °C. The mobile phases A and B were run at a flow rate of 1.0 mL/min under the gradient program of 100% B (0% A) for an initial period of 5 min, 0–20% A from 5 to 10 min, 20% A from 10 to 20 min, 20–40% A from 20 to 60 min, 40% A for 3 min, and followed by an initial step of 100% B for 5 min. PDA detection was set at 270 nm. MS analysis was done in positive ESI mode and spectra were acquired within the mass to a charge ratio (*m/z)* ranging from 100 to 700. For the single quadrupole MS system, the ESI energy was set at 70 eV, while the temperatures of the ion source and the interface were set at 150 °C and 230 °C, respectively. Nitrogen was used as the nebulizing, drying and collision gas. The capillary temperature was set to 320 °C, the nebulizer pressure was set to 60 psi and the drying gas flow rate was set to 13 L/min. Capillary voltages were set to 3500 V (positive) and 150 V (negative). The oven temperature was programmed as follows: 80 °C (held for 3 min), ramped to 110 °C at 10 °C/min (held for 5 min), increased to 190 °C (held for 3 min), ramped to 220 °C at 10 °C/min (held for 4 min) and increased to 280 °C at 15 °C/min (held for 13 min). Accurate mass measurements were performed by employing the auto mass calibration method using an external mass calibration solution (ESI-L Low Concentration Tuning Mix; Agilent calibration solution B). Herein, the limit of detection (LOD), limit of quantitation (LOQ) and recovery value were found to be 0.5 mg/kg, 1.20 mg/kg and 70–110%, respectively. The chromatographic and mass spectrometric analysis and prediction of the chemical formula, including the exact mass calculation, were performed by Mass Hunter software version B.04.00 built to 4.0.479.0 (Agilent Technology). Available authentic phenolics (1 mg/mL each), such as gallic acid, catechin, tannic acid, rutin, isoquercetin, hydroquinine, eriodictyol and quercetin, were also analyzed and used to generate a database. In addition, MS data were searched for in published literature repositories.

### 4.4. UHPLC-ESI-QTOF-MS Analysis of Phytochemical Compounds

Phytochemical compounds of the coffee extracts were analyzed at the Central Laboratory, Faculty of Agriculture, Chiang Mai University, Chiang Mai, Thailand using the HPLC-ESI-QTOF-MS method [88]. The HPLC instrument was equipped with an ESI-QTOF-MS machine (Agilent Acquisition SW version 6200 series TOF/6545 series LC/Q-TOF) and QTOF Firmware Version 25.698 (Agilent Technologies, Santa Clara, CA, USA). Mobile phase A (acetonitrile) and mobile phase B (0.1% formic acid) were degassed at 25 °C for 15 min. The GSO (20 mg) was constituted in 1.0 mL of the A:B mixture (1:1, *v/v*). It was then filtered using a syringe filer (polyvinylidene fluoride type, 0.45 µm pore size, Millipore, MA, USA) and put into HPLC vials. The flow rate was set to 0.35 mL/min, the injection volume was measured at 10 µL for each sample and the running time was 60 min. Chromatographic separation was carried out on a column (InfinityLabPoroshell 120 EC-C18 type, 2.1 mm × 100 mm, 2.7 µm, Agilent Technologies, Santa Clara, CA, USA) that was regulated thermally at 40 °C. The ESI-MS-MS spectra were recorded using an Agilent Q-TOF mass spectrometer. In the MS system, nitrogen gas nebulization was set at 45 pounds per inch^2^ with a flow rate of 5 L/min at 300 °C, while the sheath gas was set at 11 L/min at 250 °C. In addition, the capillary and nozzle voltage values were set at 3.5 kV and 500 V, respectively. A complete mass scan was conducted as a mass-to-charge ratio (*m/z*) ranging from 200 to 3200. All the operations, acquisition and analysis of the data were monitored using Agilent LC-Q-TOF-MS MassHunter Acquisition Software Version B.04.00 (Agilent Technologies) and operated under MassHunter Qualitative Analysis Software Version B.04.00 (Agilent Technologies). Peak identification was performed in positive modes using the library database, while the identification scores were further selected for characterization and *m/z* verification.

### 4.5. HPLC-DAD Analysis of Caffeic Acid, Caffeine and Chlorogenic acid Contents

Owing to their popularity and the general preferences of consumers, we focused our investigation on the biological properties of roasted coffee extracts in comparison with instant coffee. Thus, CA, CGA and CF contents were analyzed in boiled, drip and espresso roasted coffee extracts using the HPLC-DAD method [65]. The conditions included a column (C18-type, 4.6 × 250 mm, 5-µm particle size, Agilent Technologies, Santa Clara, California, United States), isocratic mobile-phase solvent containing 0.2% phosphoric acid and acetonitrile (90:10, *v/v*), a flow rate of 1.0 mL/min and a wavelength detection of 275 nm for CF and 330 nm for CA and CGA. Data were recorded and integrated using Millenium 32 HPLC Software. CGA and CF were identified by comparison with the specific T_R_ of the authentic standards. Determined concentrations were then established from the standard curves constructed from different concentrations.

### 4.6. Determination of Total Phenolic Content

Total phenolic compounds (TPC) of the roasted coffee extracts and instant coffee were determined using the Folin-Ciocalteu method [89]. Briefly, coffee samples (100 µL) were incubated with 10% Folin-Ciocalteu reagent (200 µL) at room temperature for 4 min and then incubated with 700 mM Na_2_CO_3_ (800 µL) for 30 min. The optical density (OD) was then measured at 765 nm against the reagent blank. GA (6.25–200 µg/mL) was used to generate a calibration curve that was then used to calculate TPC and identify the gallic acid equivalent (GAE).

### 4.7. Determination of Free-Radical Scavenging Activity

Antioxidant activity was determined by using the ABTS radical cation (ABTS^●+^) decolorization method [89] and the DPPH radical (DPPH^●^) scavenging method as was previously established by Hatano and colleagues with slight modifications [89]. Ten microliters of DI (control), coffee extracts (50–4000 µg/mL), instant coffee (50–4000μg/mL) or Trolox (6.25–800 µg/mL) was incubated with the freshly prepared solution consisting of 3.5 mM ABTS^●+^ and 1.22 mM K_2_S_s_O_8_ in the dark at room temperature for exactly 6 min. The substance was then photometrically measured at a wavelength of 764 nm. Results are expressed as percentage of inhibition of ABTS^●+^ production and reported in mg Trolox equivalent antioxidant capacity as (TEAC)/g dry weight of the extracts. With the use of the DPPH^●^ scavenging method, DI (control), coffee extracts (30–500 μg/mL), instant coffee (30–500μg/mL) or Trolox (6.25–50 µg/mL) were incubated with of 0.4 mM DPPH^●^ solution in the dark at room temperature for 30 min and photometrically measured at a wavelength of 517 nm. Antioxidant activity was determined using the following equation:Radical scavenging activity (%) = [(OD_DI_ − OD_coffee or Trolox_)/OD_DIt_] × 100(1)

Results were reported as percentage of ABTS^●^ or DPPH^●^ scavenging activity (*y*-axis) and compared to DI plotted against coffee extract, instant coffee or Trolox concentration (*x*-axis), while the concentration of the substance that provided 50% reduction of DPPH^●^ concentration (IC_50_) value was calculated from the graph.

### 4.8. Determination of Iron-Binding Ability

Iron-binding activity of the coffee extracts and the instant coffee was determined with the method established by Srichairatanakool and colleagues [90]. Nitrilotriacetate (NTA) pH 7.0 solution (80 mM) was prepared by mixing 80 mM nitrilotriacetate disodium salt solution with 80 mM nitrilotriacetate trisodium salt solution until the pH reached 7.4. Stock ferric nitrilotriacetate (Fe-NTA) solution (1 mM) was freshly prepared by dissolving ferric nitrate nonahydrate in 80 mM NTA pH 7.0 solution and the mixture was incubated at room temperature for 1 h. Coffee extracts (250 µg/mL) were diluted in 10 mM MOPS buffer pH 7.0 to reach designated concentrations. The coffee extracts, CGA and CF, were incubated with 1 mM Fe-NTA solution at room temperature for 1 h and were photometrically measured within a wavelength range of 200–800 nm using a scanning double-beam UV-VIS spectrophotometer against the reagent blank. In addition, the coffee extracts (250 µg/mL) were incubated with different concentrations of Fe-NTA solution at room temperature for 30 min. Finally, the produced coffee-iron complex was photometrically measured within a wavelength range of 200–800 nm using a scanning double-beam UV-VIS spectrophotometer against the reagent blank.

### 4.9. Cell Toxicity Study Using MTT Assay

Colorimetric MTT assay was used for the determination of cell viability based on mitochondrial reductases presented in live cells, in which yellow MTT dye was reduced to purple-colored formazan product in a direct proportion to the number of viable cells [91]. Normal human peripheral blood mononuclear cells (PBMC) and two human breast cancer cell lines were used in the toxicity study. The blood collection protocol was approved of by the director of Maharaj Nakorn Chiang Mai, Faculty of Medicine, ChiangMai University, Chiang Mai, Thailand and the Ethical Committee for Human Study (Reference Number 8393 (8).9/436). Informed consent was provided by healthy blood volunteers. Blood samples (10 mL) obtained from healthy donors were collected into tubes (BD Vacutainer^®^, USA) containing 158 USP units of sodium heparin and diluted with sterile PBS, pH 7.4(1:1, *v/v*). Ficoll-Paque (Histopaque^®^-1077) solution was then overlaid with heparinized blood (at a ratio of 1:2, *v/v*) in 50-mL sterile conical tubes and centrifuged at 1500 rpm and at a temperature of 25 °C for 30 min. After centrifugation, the upper layer was aspirated and mononuclear cells at the interphase were collected into 15-mL sterile centrifuge tubes. The suspension was washed twice with 10 mL of sterile PBS, then centrifuged at 6000 rpm and at a temperature of 4 °C for 5 min, while the supernatant was aspirated and the cell pellets were resuspended in complete culture medium. The suspension was then incubated overnight in a humidified environment containing 5%CO_2_ at 37 °C. In the assay, PBMC (7 × 10^4^ cells/well) were incubated with boiled, drip and espresso coffee extracts (3.125–50 µg/mL) or PBS for 24 and 48 h in a humidified 5%CO_2_, 37 °C incubator. After the treatment, MTT reagent (20 µL) was added into each well and the cells were incubated in the dark at 37 °C for 4 h. Finally, the formazan crystal product was dissolved with DMSO (200 µL) and the OD was measured at wavelengths of 570 and 630 nm. The percentage of cell viability was calculated using the following formula: OD_sample_/OD_control_× 100.

The human breast cancer cell lines, MCF-7 (ATCC HTB-22) and MDA-MB-231 (ATCC HTB-26), were kindly provided by Professor Dr. Masami Suganuma, PhD. at the Graduate School of Science and Engineering, Saitama University, Saitama, Japan and Professor Dr. Ratana Banjerdpongchai, MD., PhD. at the Department of Biochemistry, Faculty of Medicine, Chiang Mai University, Thailand. MDA-MB-231 cells were cultured incomplete RPMI-1640 medium supplemented with 10% (*v*/*v*) heat-inactivated fetal bovine serum (FBS) and penicillin-streptomycin solution (10,000 U/mL). They were then treated with boiled, drip and espresso coffee extracts (3.12–50 µg/mL) for 24 and 48 h. Viability was then determined using the MTT assay. In another study, MDA-MB-231 and MCF-7 cells were treated with drip coffee extracts (31.25–1000µg/mL) in an incubator (5% CO_2_, 37 °C) for 24 and 48 h. Viability was then determined using the MTT assay.

### 4.10. Acute Toxicity Study

Acute toxicity of the drip coffee extract was investigated in female rats for 14 days using the Standard Operating Procedure (SOP) according to OECD Guidelines 2008 established by van den Heuvel and colleagues [92]. Animals were purchased from the National Laboratory Animal Center, Mahidol University, Salaya Campus, Thailand. The study protocol was approved of by the Animal Ethics Committee, Faculty of Medicine, Chiang Mai University, Thailand (Reference Number 25/2561). Male Sprague–Dawley rats at approximately 8–10 weeks of age and weighing around 180–200 g were used for the acute toxicity study. Rats were orally fed with a single dose of drip coffee extract using an oral dosing needle. Animals were observed after being dosed for 30 min, 1, 4 and 6 h and daily for 14 d. In addition, rat body weights were recorded before treatment, then once a week for the next 2 weeks or at the time of death. The initial dose was estimated from the median lethal dose (LD_50_) of caffeine content, while the starting dose was 175 mg/kg BW. The dose for the next animal was then adjusted up or down (1.3 of dose progression factor) depending upon the outcome for the first animal. If an animal survived, the dose for the next animal was increased; if the animal died, the dose for the next animal was decreased. The upper and lower limits of the dose should not exceed 2000 mg/kg BW or be lower than 5 mg/kg BW, respectively. Remark: The caffeine content in the upper limit dose should not exceed 200 mg/kg BW. Visual observations including behavioral patterns (trembling, diarrhea, breathing, impairment in food intake, water consumption, postural abnormalities, hair loss, sleep, lethargy and restlessness) and physical appearance (eye color, mucous membrane, salivation, skin/fur effects, body weight and injury) were recorded daily for 14 d.

### 4.11. Statistical Analysis

The experiments were performed in at least triplicate. Data were analyzed using the SPSS Statistics Program (IBM SPSS^®^ Software version 22, IBM Corporation, Armonk, NY, USA, with a shared license assigned to Chiang Mai University, Thailand) and are expressed as mean±standard deviation (SD) or mean±standard error of the mean (SEM) values. Statistical significance was analyzed using one-way analysis of variance (ANOVA) followed by Tukey HSD’s post-test. A *p*-value < 0.05 was considered significant.

## 5. Conclusions

We have conducted a comparative analysis of the phytochemical compounds, radical scavenging activities and also the iron binding capacity of roasted and green Arabica coffee extracts and instant coffee. The profiles of the phenolics and phytochemicals were practically the same in all green and roasted coffee bean extracts prepared using the boiling, drip-filtered and espresso methods. An exception of this would be the existence of additional dimethoxycinnamic acid, caffeoylarbutin and cynaroside that were present in green coffee. Similar to instant coffee, besides an abundance of caffeine, all the roasted coffee extracts were determined to be rich phenolic compounds; particularly, chlorogenic acid, which is known to exert dose dependent antioxidant and iron-binding properties. Among them, the drip coffee extract contained higher total phenolic content which would reflect stronger free-radical scavenging activity. The coffee extracts were not harmful to breast cancer cells and normal white blood cells, while they did support viability of the white blood cells at longer incubation times. Taken together, an automatic drip-filter coffee maker would be the most efficient way of brewing roasted coffee to yield the highest free-radical capacity, as well as the greatest amounts of phenolic compounds.

## Figures and Tables

**Figure 1 molecules-26-04169-f001:**
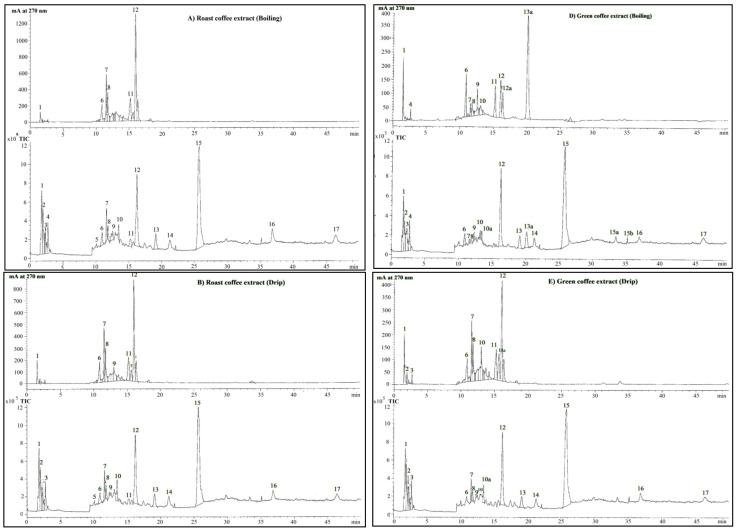
Chromatograms of the phenolic compounds in roasted and green coffee extracts. Roasted and green coffee were extracted with hot water using the boiling, drip and espresso methods. Phenolic compounds in the coffee extracts were identified using HPLC-ESI-MS, as has been described in Section 4.3.

**Figure 2 molecules-26-04169-f002:**
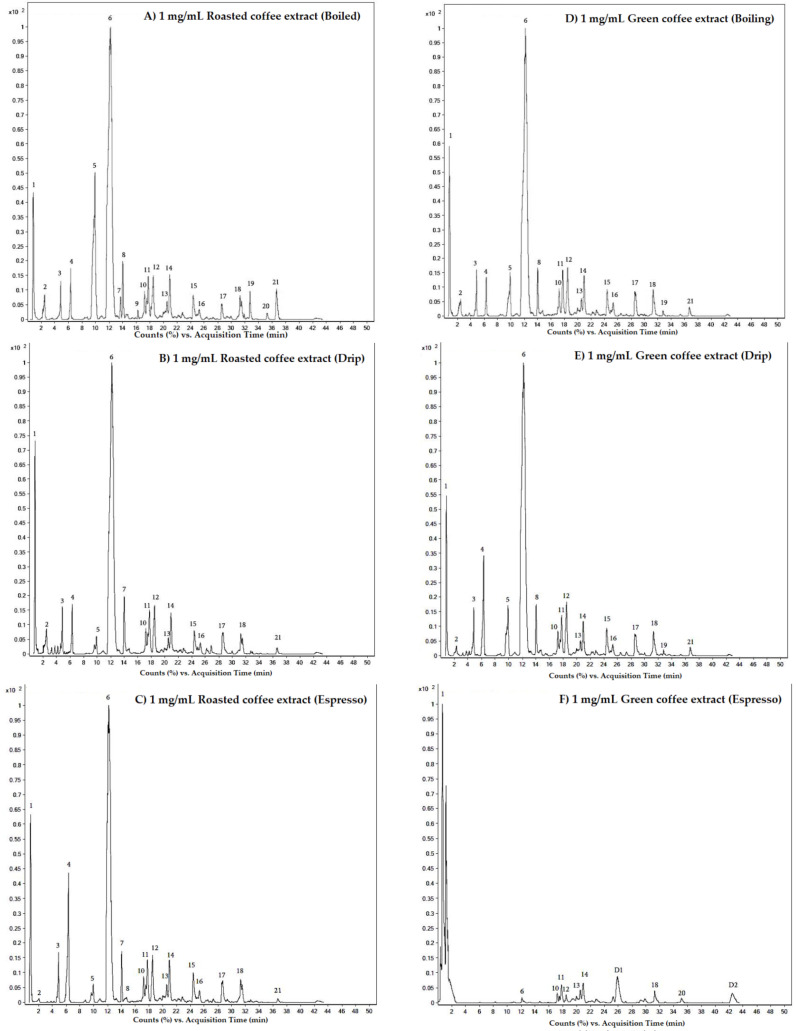
Chromatograms of the phytochemical compounds in roasted and green coffee extracts. Extracts of roasted coffee (**A**–**C**) and green coffee (**D**–**F**) were prepared by the boiling, drip and espresso methods, respectively. The coffee extracts were then identified using UHPLC-ESI-QTOF-MS, as has been described in Section 4.5.

**Figure 3 molecules-26-04169-f003:**
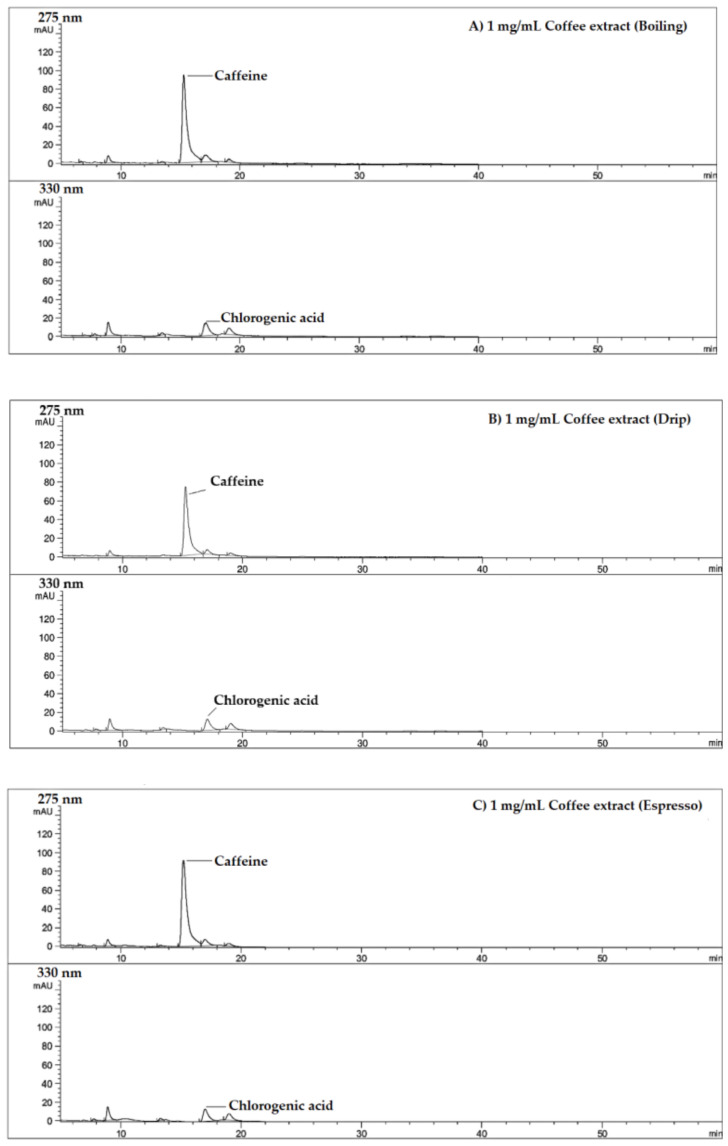
Chromatograms of coffee extracts, standard caffeic acid, caffeine and chlorogenic acid. Extracts of roasted coffee were prepared by the boiling, drip and espresso methods ((**A**–**C**), respectively). The coffee extracts and authentic standards, including caffeic acid, caffeine and chlorogenic acid, were then subjected to HPLC-DAD analysis as has been described in Section 4.5.

**Figure 4 molecules-26-04169-f004:**
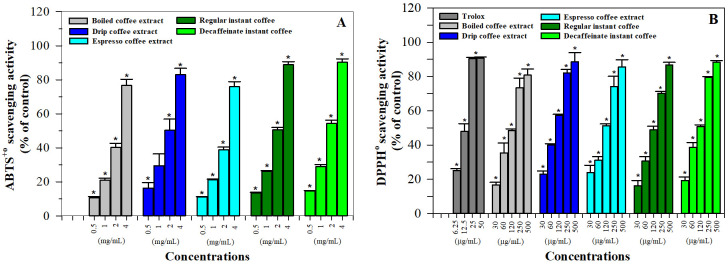
Free-radical scavenging activity of roasted coffee extracts and instant coffee. (**A**) ABTS^+●^ solution was incubated with deionized water (DI), standard Trolox, boiled, drip and espresso roasted coffee extracts, as well as regular and decaffeinated instant coffee products, while optical density (OD) was photometrically measured at a wavelength of 764 nm against the reagent blank. (**B**) DPPH^●^ solution was incubated at various concentrations of standard Trolox, boiled, drip and espresso coffee extracts, regular and decaffeinated instant coffee products and OD was photometrically measured at a wavelength of 517 nm against the reagent blank. Radical-scavenging activity was calculated according to the method described in Section 4.7 and is reported as percentage of ABTS^+●^ or DPPH^●^ scavenging activity. Data obtained from three repetitions are expressed as mean ± SD values.

**Figure 5 molecules-26-04169-f005:**
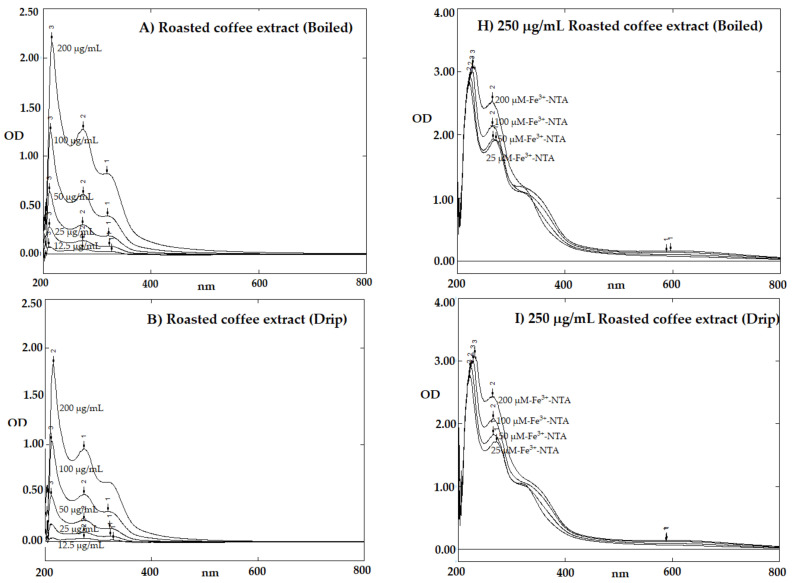
Spectral patterns of roasted coffee extracts, instant coffee, CF and CGA with and without the addition of iron. (**A**–**G**) Boiled, drip and espresso roasted coffee extracts (25–200 μg/mL), regular and decaffeinated instant coffee (12.5–200 µg/mL each), CF (2.45–38.8 µg/mL) and CGA (2.22–35.43 µg/mL) were photometrically measured at a wavelength rank of 200–800 nm against the reagent blank. (**H**–**N**) The coffee extracts and the instant coffees (250 μg/mL), CF (500 μM) and CGA (250 μM) were incubated with Fe-NTA (25–200 µM) for 1 h and photometrically measured at a wavelength rank of 200–800 nm against the coffee extract, instant coffee, CF and CGA solutions. Abbreviations: CF = caffeine, CGA = chlorogenic acid, Fe-NTA = ferric nitrilotriacetic acid.

**Figure 6 molecules-26-04169-f006:**
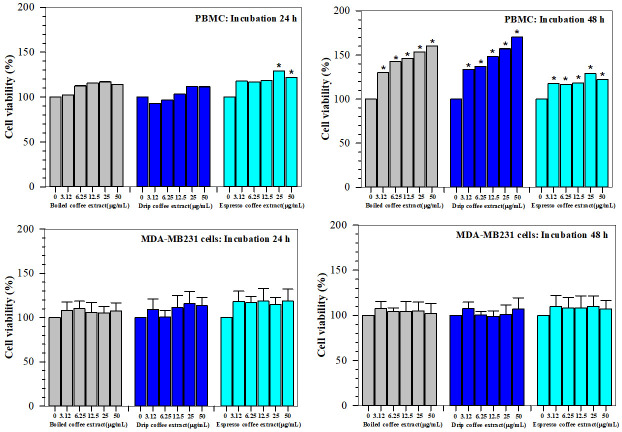
Viability of PBMC, MDA-MB-231 and MCF-7 cells treated with roasted coffee extracts. Normal human peripheral blood mononuclear cells (PBMC) and human breast cancer (MDA-MB-231) cell lines were treated with boiled, drip and espresso roasted coffee extracts for 24 and 48 h. The degree of viability was determined using MTT assay. Using aggressive doses, MDA-MB-231 and MCF-7 cells were treated with drip roasted coffee extract and the degree of viability was determined using MTT assay. Data are expressed as mean ± SEM values. * *p* < 0.05 when compared without treatment.

**Table 1 molecules-26-04169-t001:** Description of all coffee samples used in the experiments.

Coffee Samples	Source	Preparation	Condition/Instrument
Roasted Arabica coffee beans	Royal Project Foundation, Thailand	Extraction	Boiled water
		Extraction	Automatic drip coffee machine
		Extraction	Portable espresso coffee machine
Green Arabica coffee beans	Royal Project Foundation, Thailand	Extraction	Boiled water
		Extraction	Automatic drip coffee machine
		Extraction	Portable espresso coffee machine
Regular instant coffee	Tesco Supermarket, Thailand	Brewing	nd
Decaffeinated instant coffee	Tesco Supermarket, Thailand	Brewing	nd

nd = not determined.

**Table 2 molecules-26-04169-t002:** Identification of phenolic compounds in roasted and green coffee extracts. Roasted and green coffee were extracted using the boiling, drip and espresso methods. Phenolic compounds in the coffee extracts were identified using HPLC-ESI-MS, as has been described in Section 4.3.

**(A) Roasted Coffee Extract (Boiling)**										
**Peak**	**OD 270 nm**	**TIC MS**	**Exact Mass**	**Chemical**	**Observed Mass (*m*/*z*)**				**Error**	**Identification**
**No**	**T_R_ (min)**	**T_R_ (min)**	**(g/mol)**	**Formula**	**[M-H]^+^**	**[M-NH_4_]^+^**	**[M-Na]^+^**	**[M-K]^+^**	**(%)**	
1	1.52	1.77	218.1	NA	219.1	231.1	-	262.2	0.46	Unknown
2	1.91	1.95	137.1	C₇H₇NO₂	138.1	154.1	-	176.1	0.73	Trigonelline
3	2.65	2.69	180.2	C_9_H_8_O_4_	182.2	194.2	203.2	220.1	1.13	CA
4	ND	3.07	170.1	C_7_H_6_O_5_	171.1	183.1	193.1	202.2	0.58	GA
5	10.28	10.09	290.3	C_15_H_14_O_6_	291.1	303.0	314.1	325.1	0.28	Epicatechin
6	10.90	10.94	516.4	C_25_H_24_O_12_	517.1	527.0	538.0	554.1	0.14	DiCGA
7	11.56	11.60	354.3	C_16_H_18_O_9_	355.0	367.2	377.0	394.1	0.20	CGA
8	11.78	11.83	342.3	C_15_H_18_O_9_	343.0	355.1	365.0	381.1	0.20	Caffeoyl-*O*-hexoside
9	12.49	12.54	338.3	C_16_H_18_O_8_	339.1	-	361.1	377.1	0.23	*p*-Coumaroylquinic acid
10	13.71	13.45	327.3	C_15_H_18_O_8_	328.2	341.1	351.2	-	0.27	*p*-Coumaroyl glycoside
11	15.22	15.25	610.5	C_27_H_30_O_16_	611.1	-	633.0	649.0	0.10	Rutin
12	16.01	16.23	452.0	C_22_H_28_O_10_	453.3	466.0	476.2	491.2	0.29	Unknown CGA derivative
13	ND	19.10	198.2	C_9_H_10_O_5_	198.1	214	227.2	235.2	−0.05	Syringic acid
14	ND	21.23	516.1	C_22_H_28_O_14_	517.0	529.0	-	547.0	0.17	CGA-*O*-glucoside
15	ND	25.90	678.1	C_28_H_39_O_21_	679.4	-	-	-	0.19	CGAdiglucoside I
16	ND	36.79	682.0	C_30_H_34_O_17_	680.4	-	-	-	−0.23	CGA diglucoside II
17	ND	46.33	682.0	C_30_H_34_O_17_	680.4	-	-	-	−0.23	DiCGA*O*-glucoside
**(B) Roasted Coffee Extract (Drip)**										
**Peak**	**OD 270 nm**	**TIC MS**	**Exact Mass**	**Chemical**	**Observed Mass (*m*/*z*)**				**Error**	**Identification**
**No**	**T_R_ (min)**	**T_R_ (min)**	**(g/mol)**	**Formula**	**[M-H]^+^**	**[M-NH_4_]^+^**	**[M-Na]^+^**	**[M-K]^+^**	**(%)**	
1	1.50	1.77	218.1	NA	219.1	231.1	-	262.2	0.46	Unknown
2	1.91	1.95	137.1	C₇H₇NO₂	138.1	154.1	-	176.1	0.73	Trigonelline
3	2.65	2.70	180.2	C_9_H_8_O_4_	182.2	194.2	206.1	219.1	1.13	CA
6	10.90	10.93	516.4	C_25_H_24_O_12_	517.1	527.2	538.1	555.0	0.14	DiCGA
7	11.56	11.60	354.3	C_16_H_18_O_9_	355.1	367.2	377.1	394.1	0.23	CGA
8	11.78	11.83	342.3	C_15_H_18_O_9_	344.1	355.1	365.0	381.1	0.53	Caffeoyl-*O*-hexoside
9	12.59	12.54	338.3	C_16_H_18_O_8_	339.1	344.2	365.1	377.1	0.23	*p*-Coumaroylquinic acid
10	13.71	13.45	327.3	C_15_H_18_O_8_	328.2	342.1	355.1	-	0.27	*p*-Coumaroyl glycoside
11	15.22	15.25	610.5	C_27_H_30_O_16_	611.0	-	633.0	649.0	0.08	Rutin
12	16.02	16.21	452.0	C_22_H_28_O_10_	453.3	465.0	476.1	491.2	0.29	Unknown CGA derivative
13	ND	19.12	198.2	C_9_H_10_O_5_	198.1	214.1	227.2	235.1	−0.05	Syringic acid
14	ND	21.23	516.1	C_22_H_28_O_14_	517.0	529.1	-	545.0	0.17	CGA-*O*-glucoside
15	ND	25.67	678.1	C_28_H_39_O_21_	679.4	-	-	-	0.19	CGA diglucoside I
16	ND	36.83	682.0	C_30_H_34_O_17_	680.4	-	-	-	−0.23	CGA diglucoside II
17	ND	46.33	682.0	C_30_H_34_O_17_	680.4	-	-	-	−0.23	DiCGA-*O*-glucoside
**(C) Roasted Coffee Extract (Espresso)**										
**Peak**	**OD 270 nm**	**TIC MS**	**Exact Mass**	**Chemical**	**Observed Mass (*m*/*z*)**				**Error**	**Identification**
**No**	**T_R_ (min)**	**T_R_ (min)**	**(g/mol)**	**Formula**	**[M-H]^+^**	**[M-NH_4_]^+^**	**[M-Na]^+^**	**[M-K]^+^**	**(%)**	
1	1.77	1.77	218.1	NA	219.1	-	-	262.1	0.46	Unknown
2	1.84	1.95	137.1	C₇H₇NO₂	138.1	154.1	-	176.1	0.73	Trigonelline
3	2.70	2.70	180.2	C_9_H_8_O_4_	182.1	194.1	206.2	219.1	1.08	CA
6	10.91	10.95	516.4	C_25_H_24_O_12_	517.1	-	-	-	0.14	DiCGA
7	11.58	11.62	354.3	C_16_H_18_O_9_	355.0	367.2	377.0	394.1	0.20	CGA
8	11.88	11.85	354.3	C_16_H_18_O_9_	355.1	367.1	377.1	393.0	0.23	Caffeoyl-*O*-hexoside
9	12.60	12.33	442.4	C_22_H_18_O_10_	445.1	457.1	-	487.0	0.61	Epicatechin 3-gallate
10	13.08	13.22	313.1	C_15_H_18_O_8_	314.1	327.1	337.0	347.1	0.32	*p*-Coumaroyl glycoside
11	15.26	15.30	610.5	C_27_H_30_O_16_	611.0	-	633.0	649.0	0.08	Rutin
12	16.06	16.25	452.0	C_22_H_28_O_10_	453.3	465.0	475.2	491.1	0.29	Unknown CGA derivative
13	ND	19.13	198.2	C_9_H_10_O_5_	198.1	214.1	227.2	235.1	−0.05	Syringic acid
14	ND	21.23	516.1	C_22_H_28_O_14_	517.0	529.1	-	545.0	0.17	CGA-*O*-glycoside
15	ND	25.76	678.1	C_28_H_39_O_21_	679.4	-	-	-	0.19	CGA diglucoside I
16	ND	36.83	682.0	C_30_H_34_O_17_	680.4	-	-	-	−0.23	CGA diglucoside II
17	ND	46.33	682.0	C_30_H_34_O_17_	680.4	-	-	-	−0.23	DiCGA-*O*-glucoside
**(D) Green Coffee Extract (Boiling)**										
**Peak**	**OD 270 nm**	**TIC MS**	**Exact Mass**	**Chemical**	**Observed Mass (*m*/*z*)**				**Error**	**Identification**
**No**	**T_R_ (min)**	**T_R_ (min)**	**(g/mol)**	**Formula**	**[M-H]^+^**	**[M-NH_4_]^+^**	**[M-Na]^+^**	**[M-K]^+^**	**(%)**	
1	1.50	1.77	218.1	NA	219.1	-	-	262.1	0.46	Unknown
2	1.91	1.95	137.1	C₇H₇NO₂	138.1	154.1	-	176.1	0.73	Trigonelline
3	2.66	2.70	180.2	C_9_H_8_O_4_	182.1	194.1	206.2	219.1	1.08	CA
6	10.91	10.95	392.0	C_25_H_24_O_12_	393.0	405.1	401.1	431.0	0.26	DiCGA
7	11.22	11.61	354.3	C_16_H_18_O_9_	355.0	367.2	377.0	394.1	0.20	CGA
8	11.67	11.84	342.3	C_15_H_18_O_9_	343.0	355.1	365.0	381.1	0.20	Caffeoyl-O-hexoside
9	12.09	12.32	442.4	C_22_H_18_O_10_	445.1	457.1	-	487.0	0.61	Epicatechin 3-gallate
10	13.08	13.22	313.1	C_15_H_18_O_8_	314.1	328.3	337.0	347.1	0.32	*p*-Coumaroyl glycoside
10a	13.46	13.47	208.2	C_11_H_12_O_4_	*207.1*	-	-	-	−0.53	Dimethoxycinnamic acid
11	15.26	ND	610.5	C_27_H_30_O_16_	611.1	-	633.0	649.0	0.10	Rutin
11a	15.64	ND	434.4	C_21_H_22_O_10_	*433.0*	-	-	-	−0.32	Caffeoylarbutin
12	16.07	16.25	452.0	C_22_H_28_O_10_	453.3	466.0	476.2	491.2	0.29	Unknown CGA derivative
13	ND	19.14	198.2	C_9_H_10_O_5_	198.1	214	227.2	235.2	−0.05	Syringic acid
14	ND	21.29	516.1	C_22_H_28_O_14_	517.0	529.1	-	545.0	0.17	CGA-*O*-glucoside
15	ND	25.75	678.1	C_28_H_39_O_21_	679.4	-	-	-	0.19	CGA diglucoside I
16	ND	36.86	682.0	C_30_H_34_O_17_	680.4	-	-	-	−0.23	CGA diglucoside II
17	ND	46.41	682.0	C_30_H_34_O_17_	680.4	-	-	-	−0.23	DiCGA-*O*-glucoside
**(E) Green Coffee Extract (Drip)**										
**Peak**	**OD 270 nm**	**TIC MS**	**Exact Mass**	**Chemical**	**Observed Mass (*m*/*z*)**				**Error**	**Identification**
**No**	**T_R_ (min)**	**T_R_ (min)**	**(g/mol)**	**Formula**	**[M-H]^+^**	**[M-NH_4_]^+^**	**[M-Na]^+^**	**[M-K]^+^**	**(%)**	
1	1.05	1.77	218.1	-	219.1	231.1	-	262.2	0.46	Unknown
2	1.91	1.95	137.1	C₇H₇NO₂	138.1	154.1	-	176.1	0.73	Trigonelline
3	2.65	2.70	180.2	C_9_H_8_O_4_	182.2	194.2	203.2	220.1	1.13	CA
6	10.92	10.92	516.4	C_25_H_24_O_12_	517.1	527.0	538.0	554.1	0.14	DiCGA
7	11.57	11.61	354.3	C_16_H_18_O_9_	355.0	367.2	377.0	394.1	0.20	CGA
8	11.79	11.84	342.3	C_15_H_18_O_9_	343.0	355.1	365.0	381.1	0.20	Caffeoyl-*O*-hexoside
9	ND	12.32	338.3	C_16_H_18_O_8_	339.1	-	361.1	377.1	0.23	*p*-Coumaroylquinic acid
10	13.01	ND	313.1	C_15_H_18_O_8_	314.1	328.3	337.0	347.1	0.32	*p*-Coumaroylglycoside
10a	13.42	13.46	208.2	C_11_H_12_O_4_	*207.1*	-	-	-	−0.53	Dimethoxycinnamic acid
11	15.14	ND	610.5	C_27_H_30_O_16_	611.1	-	633.0	649.0	0.10	Rutin
11a	15.63	15.67	434.4	C_21_H_22_O_10_	*433.0*	-	-	-	−0.32	Caffeoylarbutin
12	16.05	16.23	452.0	C_22_H_28_O_10_	453.3	466.0	476.2	491.2	0.29	Unknown CGA derivative
13	ND	19.11	198.2	C_9_H_10_O_5_	198.1	214	227.2	235.2	−0.05	Syringic acid
14	ND	21.29	516.1	C_22_H_28_O_14_	517.0	529.1	-	545.0	0.17	CGA-*O*-glucoside
15	ND	25.79	678.1	C_28_H_39_O_21_	679.4	-	-	-	0.19	CGA diglucoside I
16	ND	36.64	682.0	C_30_H_34_O_17_	680.4	-	-	-	−0.23	CGA diglucoside II
17	ND	46.40	682.0	C_30_H_34_O_17_	680.4	-	-	-	−0.23	DiCGA-*O*-glucoside
**(F) Green Coffee Extract (Espresso)**										
**Peak**	**OD 270 nm**	**TIC MS**	**Exact Mass**	**Chemical**	**Observed Mass (*m*/*z*)**				**Error**	**Identification**
**No**	**T_R_ (min)**	**T_R_ (min)**	**(g/mol)**	**formula**	**[M-H]^+^**	**[M-NH_4_]^+^**	**[M-Na]^+^**	**[M-K]^+^**	**(%)**	
1	1.50	1.77	218.1	NA	219.1	231.1	-	262.2	0.46	Unknown
2	ND	1.95	137.1	C₇H₇NO₂	138.1	154.1	-	176.1	0.73	Trigonelline
3	2.65	2.69	180.2	C_9_H_8_O_4_	182.2	194.2	203.2	220.1	1.13	CA
6	10.90	10.94	516.4	C_25_H_24_O_12_	517.1	527.0	538.0	554.1	0.14	DiCGA
7	11.56	11.61	354.3	C_16_H_18_O_9_	355.0	367.2	377.0	394.1	0.20	CGA
8	11.79	11.84	342.3	C_15_H_18_O_9_	343.0	355.1	365.0	381.1	0.20	Caffeoyl-*O*-hexoside
9	12.59	12.32	338.3	C_16_H_18_O_8_	339.1	-	361.1	377.1	0.23	*p*-Coumaroylquinic acid
10	ND	13.21	313.1	C_15_H_18_O_8_	314.1	328.3	337.0	347.1	0.32	*p*-Coumaroyl glycoside
10a	ND	13.46	208.2	C_11_H_12_O_4_	*207.1*	-	-	-	−0.53	Dimethoxycinnamic acid
11	15.24	ND	610.5	C_27_H_30_O_16_	611.1	-	633.0	649.0	0.10	Rutin
11a	ND	15.59	302.2	C_15_H_10_O_7_	*301.1*	-	-	-	−0.38	Quercetin
12	16.04	16.23	452.0	C_22_H_28_O_10_	453.3	466.0	476.2	491.2	0.29	Unknown CGA derivative
13	ND	19.11	198.2	C_9_H_10_O_5_	198.1	214	227.2	235.2	−0.05	Syringic acid
13a	20.05	20.12	448.4	C_21_H_20_O_11_	449.0	-	471.1	487.0	0.13	Cynarosideor Luteolin-7-*O*-glucoside
14	ND	21.26	516.1	C_22_H_28_O_14_	517.0	529.1	-	545.0	0.17	CGA-*O*-glucoside
15	ND	25.67	678.1	C_28_H_39_O_21_	679.4	-	-	-	0.19	CGA diglucoside I
16	ND	36.64	682.0	C_30_H_34_O_17_	680.4	-	-	-	−0.23	CGA diglucoside II
17	ND	46.39	682.0	C_30_H_34_O_17_	680.4	-	-	-	−0.23	DiCGA-*O*-glucoside

Abbreviations: CA = caffeic acid, CGA = chlorogenic acid, DiCGA = dichlorogenic acid, GA = gallic acid, *m/z* = mass to charge ratio, MS = mass spectrometry, NA = not available, ND = not detectable, TIC = total ion count, T_R_ = retention time.

**Table 3 molecules-26-04169-t003:** Identification of phytochemical compounds in roasted and green coffee extracts. Extracts of roasted coffee and green coffee were prepared by the boiling, drip and espresso methods (A, B, C, D, E and F, respectively).The coffee extracts were then identified using UHPLC-ESI-Q-TOF-MS, as has been described in Section 4.4.

Peak No	T_R_ (min)						Observed Mass [M-H]^+^ (*m/z*)	Exact Mass (g/mol)	Error (%)	Chemical Formula	Possible Constitutes
	A	B	C	D	E	F					
1	0.76	0.72	0.72	0.75	0.75	0.77	137.05	137.11	0.000	C_4_H_8_FNO_3_	4-Fluoro-L-threonine
2	2.49	2.48	ND	2.47	2.47	ND	297.08	297.3	0.001	C_20_H_11_NO_2_	3-Nitroperylene
3	4.86	4.84	4.84	4.84	4.83	ND	490.29	490.63	0.001	C_28_H_42_O_7_	Cycloeudesmane sesquiterpenoids
4	6.31	6.31	6.31	6.29	6.30	ND	678.51	678.6	0.000	C_34_H_30_O_15_	3,4,5-Tricaffeoylquinic acids
5	9.89	9.90	9.89	9.73	9.88	ND	754.25	358.5	1.104	C_17_H_26_O_6_S	6-Gingesulfonic acid
6	12.06	12.06	12.06	12.07	12.05	12.08	317.29	317.5	0.001	C_18_H_39_NO_3_	Phytosphingosine
7	13.69	ND	ND	ND	ND	ND	226.19	226.35	0.001	C_14_H_26_O_2_	Citronellyl butyrate
8	14.02	14.01	14.02	14.02	14.03	ND	518.29	518.6	0.001	C_30_H_38_N_4_O_4_	Sativanine B
9	16.23	ND	ND	ND	ND	ND	432.18	432.46	0.001	C_23_H_28_O_8_	2-[2-(4-Hydroxy-3-meyhoxyphenyl)ethyl] tetrahydro-6-(4,5-dihydroxy-3methoxyphenyl)-2H-pyran-4-ylacetate
10	17.21	ND	ND	17.21	17.22	17.20	281.27	281.5	0.001	C_18_H_35_NO	Dodemorph
11	17.75	ND	ND	17.77	17.77	17.81	338.25	338.5	0.001	C_20_H_34_O_4_	Sterebin E
12	18.33	18.49	18.53	18.48	18.49	18.49	204.08	204.22	0.001	C_12_H_12_O_3_	Anofinic acid
13	20.52	20.55	ND	20.55	20.96	20.52	343.5	343.25	0.001	C_22_H_33_NO_2_	Samandenone (alkaloids)
14	20.96	ND	20.95	20.95	20.95	20.97	428.31	428.6	0.001	C_24_H_44_O_6_	Sorbitan oleate
15	24.38	24.36	24.43	24.39	24.39	ND	330.28	330.5	0.001	C_19_H_38_O_4_	2-Palmitoyl glycerol
16	25.28	25.28	25.28	25.33	25.32	25.58	456.34	456.70	0.001	C_33_H_44_O	Citranaxanthin
17	28.41	28.42	28.42	28.43	28.43	28.66	358.31	358.6	0.001	C_21_H_42_O_4_	2-Stearoyl glycerol
18	31.89	31.89	31.89	31.85	31.86	31.85	596.57	596.73	0.001	C_29_H_57_O_10_P	Dodecanoic acid, 13-methoxy-1-[(phosphonooxy) methyl] 1,2-ethanediyl ester
19	32.65	ND	32.52	32.76	ND	ND	596.37	596.73	0.001	C_29_H_57_O_10_P	Dodecanoic acid, 12-methoxy-1-[(phosphono oxy)methyl]-1,2-ethanediyl ester
20	34.50	ND	ND	ND	ND	ND	585.43	ND	ND	C_33_H_57_N_6_OS	Unknown
21	36.65	ND	ND	36.84	36.81	ND	641.43	ND	ND	C_41_H_57_N_2_O_4_	Unknown

Abbreviations: *m/z* = mass to charge ratio, ND = not detectable, TIC = total ion count, T_R_ = retention time.

**Table 4 molecules-26-04169-t004:** Caffeine, chlorogenic acid and total phenolic contentsin coffee samples. Boiled, drip and espresso coffee extracts, and the authentic standards including caffeic acid (CA), caffeine (CF) and chlorogenic acid (CGA), were subjected to HPLC-DAD analysis, as has been described in Section 4.5. Total phenolic contents in the coffee extracts and instant coffee were determined using the Folin-Ciocalteu method, as has been described in Section 4.6. Data obtained from three repetitions are expressed as mean ± standard deviation (SD) values.

Coffee Samples	CGA (mg/g)	CF (mg/g)	CA (mg/g)	TPC (mg GAE/g)
Boiled roasted coffee extract	14.47 ± 0.98	65.58 ± 9.83	ND	87.9 ± 7.9
Drip roasted coffee extract	15.67 ± 0.83	55.58 ± 10.61	ND	111.6 ± 11.4
Espresso roasted coffee extract	14.97 ± 0.89	64.25 ± 11.56	ND	97.2 ± 3.9
Regular instant coffee	nd	nd	nd	115.2 ± 22.4
Decaffeinated instant coffee	nd	nd	nd	118.1 ± 6.87

Abbreviations: CA = caffeic acid, CF = caffeine, CGA = chlorogenic acid, GAE = gallic acid equivalent, nd = not done, ND = not detectable, TPC = total phenolic content.

**Table 5 molecules-26-04169-t005:** Antioxidant activity values of coffee samples accessed using ABTS^+●^ and DPPH^●^ methods.

Coffee Samples	Antioxidant Activity (mg TE/g)	
	ABTS^+●^Method	DPPH^●^Method
Roasted coffee extract (boiled)	125.8 ± 9.1	1174 ± 26
Roasted coffee extract (drip)	149.4 ± 9.2	1250 ± 38
Roasted coffee extract (espresso)	127.6 ± 3.0	1274 ± 46
Regular instant coffee	160.4 ± 4.0	2359 ± 159
Decaffeinated instant coffee	173.2 ± 9.8	2358 ± 93

ABTS^+●^ = 2,2′-azino-bis(3-ethylbenzothiazoline-6-sulfonic acid) diammonium salt radical, DPPH^●^ = 2, 2-diphenyl-1-picrylhydrazylradical, TE = Trolox equivalent.

**Table 6 molecules-26-04169-t006:** Spectral peaks from roasted coffee extracts, instant coffee, CF and CGA with and without the addition of iron.

Samples	Fe-NTA (25–200 µM)	Peak(s) (nm)
Boiled coffee extract (25–200 μg/mL)	-	217, 275, 323
Drip coffee extract (25–200 μg/mL)	-	217,275, 323
Espresso coffee extract (25–200 μg/mL)	-	217, 275, 323
Regular instant coffee (12.5–200 μg/mL)	-	217, 275, 323
Decaffeinated instant coffee (12.5–200 μg/mL)	-	217, 289, 323
CF (2.45–38.8 μg/mL)	-	217, 275
CGA (2.22–35.43 μg/mL)	-	217, 324
Boiled coffee extract (250 μg/mL)	+	595
Drip coffee extract (250 μg/mL)	+	595
Espresso coffee extract (250 μg/mL)	+	595
Regular instant coffee (250 μg/mL)	+	595
Decaffeinated instant coffee (250 μg/mL)	+	595
CF (500 μM)	+	ND
CGA (250 μM)	+	617

Abbreviations: CF = caffeine, CGA = chlorogenic acid, Fe-NTA = ferric nitrilotriacetic acid, ND = not detectable.

**Table 7 molecules-26-04169-t007:** Behavioral patterns observed in rats after single oral administration of coffee extract at doses of 175, 550 and 2000 mg/kg BW. Signs of lethargy and drowsiness were observed after 1 h of administering doses of the coffee extract.

Observations	Dose of Coffee Extract			
	175 mg/kg BW (n = 1)	550 mg/kg BW (n = 1)	2000 mg/kg BW (n = 3)	
	Days 1–14	Days 1–14	Day 1	Days 2–14
Hair falling	0/1	0/1	0/3	0/3
Ophthalmopathy	0/1	0/1	0/3	0/3
Mucous membrane	0/1	0/1	0/3	0/3
Hypersalivation	0/1	0/1	0/3	0/3
Hyperpnea	0/1	0/1	0/3	0/3
Diarrhea	0/1	0/1	0/3	0/3
Lethargy	0/1	0/1	1/3 *	0/3
Drowsiness	0/1	0/1	1/3 *	0/3
Aggressiveness	0/1	0/1	0/3	0/3
Convulsions	0/1	0/1	0/3	0/3
Tremors	0/1	0/1	0/3	0/3
Mortality	0/1	0/1	0/3	0/3

* Sigh of lethargy and drowsiness were observed after 1 h of dosing the coffee extract.

## Data Availability

Data available in a publicly accessible repository.

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
