# Peer review of "Chemical Analysis, Toxicity Study, and Free-Radical Scavenging and Iron-Binding Assays Involving Coffee (Coffea arabica) Extracts"

_molecules, 2021, doi:10.3390/molecules26144169_

Round 1

Reviewer 1 Report

The paper submitted by   Nuntouchaporn Hutachok  and co-authors is very instructive description of the preparation of the Arabica coffee bean extracts, their chemical compositions,  the relevant antioxidant and iron-chelating activities and also the toxicity  of roasted coffee preparations, coffee being one of the most useful beverage in the world.

 The article is well organized and brings all necessary information useful not only for those familiar with the matter at academic community, but also for interested industry,  dealing with recently developed methodologies. From my perspective, the paper deserves the scientific public's attention.

Although overall my evaluation is positive, I recommend some revisions before the present manuscript  can be accepted for publication.

In the discussion section the data obtained (antioxidant, chelating properties, cytotoxic) should be compared with those existing in the literature.

The authors should emphasize the novelty of the conducted study and indicate more clearly the results from the study in the conclusions.

Reviewer 2 Report

The manuscript is the continuation of a previous work by the same authors. The topic is interesting but it is a bit confusing and there are several points that should be improved. For this reason, I suggest major revisions before accepting the paper for publication.

Authors should explain why not all samples were subjected to the same analyzes.

The formatting is bad and makes the article more difficult to read.

Abstract: please use the same character dimension thought the abstract;

The title of the paragraph 2.1 does not reflect its content. It is reported the extraction yield but not the amounts of CF and CGA. These last values are present but they are referred to a previous work. 

Paragraph 2.1 this is a result section only; authors should avoid reporting data from previous papers see lines 88-95

Lines 97-98. This has been already reported at the end of the introduction see lines 80-83

Line 109 leave a space between qualitative and profiles;

In table 1 the abbreviations at the end of each table section are always the same, I suggest delating and leaving only one at the end;

Table 1 section F was repeated twice,

Revise table 1 heading there are some repetitions,

Please check that in pdf file page numbering is not consequential,

Figure 4. Uniform the graphs;

I could not find table 2;

Line 171 “…were found to present in the beans..” please check English;

Paragraph 2.4 in lines 175-176 authors said that the antioxidant activity was determined with ABTS and DPPH methods and results were expressed as trolox equivalents. Later on in line 179 authors reported the results of trolox equivalents anti-oxidation, reading the text it is clear that they refer to ABTS method but it is better write it clear.

Why the authors reported ABTS results as trolox equivalents and DPPH results as EC50? Why this difference, considering that they clearly indicated that results of both methods were expressed as trolox equivalents.

Please check lines 194-203. Authors said that boiled coffee extract (187 ±9 µg/mL) exert stronger antioxidant activity than the expresso coffee extract (169±4 µg/mL) and drip coffee extract (130 ±4 µg/mL). (The same is in lines 201-202). This is not true because when you express the results as EC50 you have to keep in mind that the higher the EC50 the lower the antioxidant activity.

Check the values of EC50 of drip coffee extract, regular instant coffee and decaffeinated instant coffee.  The values written in the text does not match with the values on the graph.

Line 201: please change the world “variety”: it is a bit confusing. Reading this sentence one may think that you analyzed different coffee botanical varieties but it is not the case.

Lines 203-206: based on previous comments this sentence should be changed.

- Lines 207-208 please report in the results section results only. Moreover, it is not clear on the basis of what evidence the authors can suggest this.

- lines 229-232 please report results only

- Line 238 it is not clear which extracts has been analyzed; what the number in parentheses represents?

- line 249 the variability of these two cell lines;

- lines 255-256 based on the graphs I do not think that increasing doses of drip coffee extracts were harmful to MDA-MB-231 and MCF-7 cells

- line 265 specify BW. In general, specify abbreviations at first use;

- Lines 293-296 please check the sentence it seems that it is not finished;

- lines 301-302 the sentence is not clear;

- Please change the discussion considering the changes in the results section

- paragraph 4.2 is a bit confusing please indicate more clearly which extracts have been taken into consideration, which analyzes they underwent and how they have been prepared.

Reviewer 3 Report

The Authors presented a study dedicated to one of the most frequently dinked beverages which is coffee. Nevertheless, first general question: how many coffee samples were studies, especially in different manufacturers, and how many kinds of coffee as well as preparation techniques were used? Did you study instant coffee also? Detailed samples description (preferred in table) should be done at the beginning of the results and discussion section. Detailed comments are listed below and major revision is recommended.
l. 46 - a lot of references should be inserted here.
l. 57 "Instant coffee..." - this should be presented as a separate section.
One sentence in the introduction about UHPLC/ESI-QTOF/MS is not enough. Please add more details about previous studies in the topics connected with your studies.
Please use tables/graphs to present your results as clear as possible.
figures 1,2 quality should be enhanced - it is difficult to recognize the letters and words on graphs.
Table 1 please indicate what you mean by references.
Fig. 3 explains how the concentration was adjusted/determined and puts a description of the concentration into the figure. The units should be the same for Trolox also.
Figure 4B - the full spectra should be presented in 4B top as well as bottom part. I.e. you can add magnification if you wanted to use the same range are add an additional table with peaks maxima, etc.
Colors are free of additional charge, for that reason I m strongly recommending using the same color for the same sample type, And this should be uniformed in all text.

Round 2

Reviewer 1 Report

Thank you for revision. I accept in  this form.

Reviewer 2 Report

Authors answered to my questions but there is still one minor change they should make. In the captions of figures 1 and 3 as well as in the heading of table 2 there is no need to   indicate the unit of measurement of polyphenols  concentration considering that neither in the table nor in the figures indicated the concentration values are reported. 

Reviewer 3 Report

The Authors mostly improved the manuscript due to the Reviewers comments. Nevertheless, still, major revision is needed. Detailed comments are listed below:

  • In the abstract, as well as in the last section of the introduction clearly state what kind of samples were investigated (i.e. commercially available, selected from the local market and why did you choose these samples)
  • The presentation of references should be changed (not only at the end of the sentence but in the place where the information (data, etc.) occurred/was pointed.
  1. 49 - verify "dcaffeinated"
  2. 86 - I will say "frequently used" not "can be used"
  3. 89 coupled with
  4. 91 remove ","
  • Fig. 3 I don t see "DI" on the graph
  • Fig. 4 look at the y ax some especially where the numbers are missing
  • Fig. 5 change for bars - the presentation will be more clear

For me, one section which will summarize the obtained results and clearly presents, which samples bring the most health benefits, should be added.
